

# Pseudoscalar pair production
# via off-shell Higgs in composite Higgs models

**Diogo Buarque Franzosi[1]\*, Gabriele Ferretti[1]†, Li Huang[2]‡ and Jing Shu[3,4,5,6,7]∘**

**1** Department of Physics, Chalmers University of Technology,
Fysikgården, 41296 Göteborg, Sweden
**2** Department of Physics and Astronomy, University of Kansas,
Lawrence, Kansas, 66045 U.S.A
**3** CAS Key Laboratory of Theoretical Physics, Institute of Theoretical Physics,
Chinese Academy of Sciences, Beijing 100190, China
**4** School of Physical Sciences, University of Chinese Academy of Sciences,
Beijing 100049, P. R. China
**5** CAS Center for Excellence in Particle Physics, Beijing 100049, China
**6** Center for High Energy Physics, Peking University, Beijing 100871, China
**7** School of Fundamental Physics and Mathematical Sciences, Hangzhou Institute for
Advanced Study, University of Chinese Academy of Sciences, Hangzhou 310024, China

\* buarque@chalmers.se, † ferretti@chalmers.se, ‡ huangli@ku.edu, ∘ jshu@itp.ac.cn

## Abstract

We propose a new type of search for a pseudoscalar particle $\eta$ pair produced via an off-shell Higgs, $pp \to h^* \to \eta\eta$. The search is motivated by a composite Higgs model in which the $\eta$ is extremely narrow and decays almost exclusively into $Z\gamma$ in the mass range 65 GeV $\lesssim m_\eta \lesssim$ 160 GeV. We devise an analysis strategy to observe the novel $Z\gamma Z\gamma$ channel and estimate potential bounds on the Higgs-$\eta$ coupling. The experimental sensitivity to the signatures depends on the power to identify fake photons and on the ability to predict large photon multiplicities. This search allows us to exclude large values of the compositeness scale $f$, being thus complementary to other typical processes.



# 1 Introduction

Goldstone-Composite Higgs (CH) models are promising candidates to dynamically break the electroweak (EW) symmetry [1, 2]. They are made of three main ingredients. The first one is dynamical symmetry breaking via a condensate of new strongly interacting *hyperfermions*, which solves the hierarchy problem via dimensional transmutation. A compositeness scale $f$ is generated and identified as the decay constant of the Goldstone bosons associated with the breaking of the global symmetry [3–5]. The second ingredient is the vacuum misalignment mechanism, which creates a little hierarchy between the compositeness and EW scales $f \gg v$, and allows the identification of the Higgs boson as part of the multiplet of (pseudo-)Nambu-Goldstone bosons (pNGB), explaining its EW quantum numbers and its light nature [6]. A third, optional, ingredient is the partial compositeness (PC) mechanism [7] inducing a mass for the top quark via its mass-mixing with a fermionic operator (*aka* top partner) with large anomalous dimension in a near-conformal phase [8].

There exist several models containing these ingredients (see [9–11] for reviews), some of which also providing candidates for dark matter and addressing other pressing issues of the Standard Model (SM). Among these models, a few of them [12–17] provide the explicit matter content of hyperfermions with PC mechanism via a four-dimensional gauge theory. It is interesting and challenging to find the imprints of these hyperfermions at the Large Hadron Collider (LHC) as the direct evidence of the underlying microscopic structure.

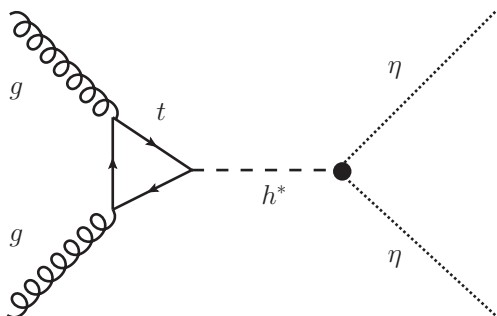

Figure 1: Main Feynman diagram contributing to $\eta$ pair production via off-shell Higgs, see (1).

One interesting common feature of all these models is the presence of an EW singlet CP odd scalar $\eta$ which is part of the same coset of pNGB as the Higgs. Its interactions are dictated by the same parameters of the Higgs sector and are constrained by CP and the non-linearly realized symmetries of the chiral lagrangian. Since it is an EW singlet, its interactions to SM

gauge bosons are mainly dictated by the Wess-Zumino-Witten (WZW) [18, 19] term and are thus suppressed. Moreover, we are interested in scenarios where its couplings to SM fermions are suppressed due to an approximate $Z_2$ symmetry $\eta \rightarrow -\eta$. This symmetry is approximately realized in specific mechanisms of fermion masses generation, either via a bilinear condensate [47] or in PC [51].

Given the fermiophobic nature of $\eta$, the diboson decay channels are the dominant ones. Interestingly, the models admitting an underlying gauge description often display a relation between the anomaly terms in the WZW interaction for which the diphoton channel vanishes *. Thus in these cases $\eta$ is both fermiophobic and photophobic, and decays predominantly to $Z\gamma$ in the mass range 65 GeV $\lesssim m_\eta \lesssim$ 160 GeV. The decays into fermion pairs, relevant for $m_\eta \lesssim$ 65 GeV, proceed via loops of gauge bosons and are further suppressed by the small anomalous couplings. However, for $m_\eta < m_h/2$, there are strong indirect bounds from the branching ratio (BR) of the Higgs into beyond-the-SM (BSM) states [20], as well as direct bounds from axion-like particle searches from exotic Higgs decays [21–30].

This leaves open an intriguing mass region $m_\eta > m_h/2$ for production at the LHC. In this mass range the leading production mode is pair production via an off-shell Higgs,

$$pp \rightarrow h^* \rightarrow \eta\eta. \tag{1}$$

The main leading order (LO) Feynman diagram contributing to this process is shown in fig. 1. Since the decay width of the $\eta$ turns out to be much smaller than the Higgs width, the $\eta$ will always be produced on-shell and the Higgs is thus forced to be off-shell.

The characteristic feature of lacking a $t-\bar{t}-\eta$ coupling suppresses the single $\eta$ production by a top loop at LHC. The top-loop box contribution is suppressed for the same reason, namely the absence of a $t-\bar{t}-\eta$ coupling. In models with PC this assertion needs to be better qualified since there could be additional couplings to top partners and contact interactions of type $\eta-\eta-t-\bar{t}$ that will eventually become relevant for large enough $\eta$ mass. This will be discussed in detail in sec. 4.

In this paper we perform a study of process (1), with $\eta$ decaying into $Z\gamma$, projecting exclusion limits to its signal strength and interpreting the results in a concrete CH model. Interestingly, this process cross section, differently from other typical processes in CH, is maximized by large compositeness scales $f$. Therefore, we obtain lower bounds in $f$, giving our analysis a complementary and novel role in the CH searches.

The paper is organized as follows. In sec. 2 we define the signal process and the simplified model used to describe it. We then discuss the simulations performed and the matching procedure to combine different photon multiplicities without double counting. In sec. 3 we present the detailed analysis, describing the selection cuts to enhance the signal and suppress the background, discussing the effect of fake photons, and providing exclusion bounds on the $\eta$-Higgs coupling. In sec. 4 we present details of a model of PC that predicts the signature of interest and interpret the exclusion bounds in terms of the compositeness scale for different top partner representations. Moreover, we discuss other production mechanisms that might compete with the off-shell Higgs, namely $\eta$-pair production via top-loop contact interaction and Vector Boson Fusion (VBF). We offer our conclusions in sec. 5.

## 2 Signal definition and simulation setup

We consider the production of a $\eta$ pair via an off-shell Higgs $h^*$, with $\eta$ decaying into a pair of fermions and a photon $\eta \rightarrow f\bar{f}\gamma$ via either an on-shell $Z$ or off-shell $Z^*$ boson. The full

---

*See Ref. [15] for explicit examples of this cancellation. Also notice that the top-loop induced coupling vanishes due to the small fermion couplings (discussed in sec. 4). The gluon decay channel is absent for the same reason.

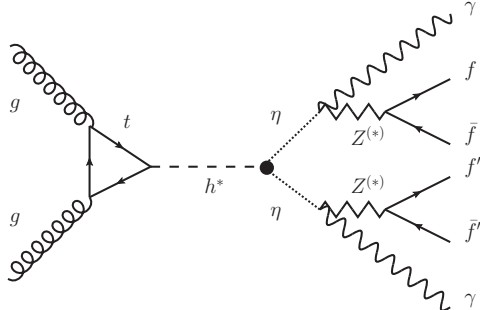

Figure 2: Diagram contributing to process (2), including the $\eta$ decay.

process

$$pp \rightarrow h^* \rightarrow \eta\eta \rightarrow Z^{(*)}\gamma Z^{(*)}\gamma \rightarrow f\bar{f}f'\bar{f}'\gamma\gamma\,, \tag{2}$$

is depicted in fig. 2 where $f, f'$ are any SM fermions.

To describe this process we adopt a photophobic and fermiophobic Lagrangian describing $\eta$ interactions,

$$\mathcal{L}_\eta = -\frac{1}{2}\lambda_\eta \frac{m_h^2}{v} h\eta^2 + \frac{\kappa}{16\pi^2 v}\eta\left(\frac{g^2 - g'^2}{2}Z_{\mu\nu}\tilde{Z}^{\mu\nu} + gg'F_{\mu\nu}\tilde{Z}^{\mu\nu} + g^2 W_{\mu\nu}^+ \widetilde{W}^{-\mu\nu}\right), \tag{3}$$

where $v = 246$ GeV is the electroweak scale, $m_h = 125$ GeV is the Higgs boson mass, $g, g'$ are the usual EW coupling constants and $\kappa$ and $\lambda_\eta$ are dimensionless quantities. This Lagrangian is well motivated by the constructions of CH models via underlying gauge theories [12–15] where the coefficients of the WZW terms can be explicitly computed and typically lead to the photophobic combination above (for instance in models based on $SU(4)/Sp(4)$ and $SU(4)\times SU(4)/SU(4)$ cosets [15,31]) in terms of a unique $\kappa = \mathcal{O}(v/f)$, suppressed by a compositeness scale $f \gtrsim 800$ GeV, while $\lambda_\eta = \mathcal{O}(1)$ is an order unity quantity. (More details are presented in sec. 4.)

We are interested in the mass region $m_h/2 < m_\eta < 150$ GeV. The $\eta$ branching ratios (BR) are shown in the left panel of fig. 3, justifying our choice of $Z^{(*)}\gamma$ as the leading channel. Despite its fermiophobic nature, loops of gauge bosons induce fermionic decays which eventually overcome the tree-level $Z^*\gamma$ decay for masses below $\approx 60$ GeV. The calculation of loop induced decays of axion-like particles is given in [32,33].

Due to the smallness of $\kappa/(16\pi^2)$ in eq. (3), $\eta$ is very narrow (eV $\lesssim \Gamma_\eta \lesssim$ keV in the motivated mass region), and can safely be assumed to be produced on its mass shell. In particular we can use the narrow width approximation and the cross section of process (2) can be factorized as

$$\sigma = \lambda^2 \sigma_0 BR(\eta \rightarrow Z^{(*)}\gamma)^2 BR(Z^{(*)} \rightarrow f\bar{f}) BR(Z^{(*)} \rightarrow f'\bar{f}')\,, \tag{4}$$

with

$$\lambda \equiv \lambda_\eta \kappa_t\,, \tag{5}$$

where $\kappa_t$ is the deviation of the Higgs coupling to $t\bar{t}$ w.r.t. the SM value and $\sigma_0$ is the production cross section of $(pp \rightarrow \eta\eta)$ with $\lambda = 1$. Of course, for $Z$ off-shell the factorization $BR(\eta \rightarrow Z^*\gamma)BR(Z^* \rightarrow f\bar{f})$ is meaningless, but we consider it only as a short-hand for the $\eta$ three-body decay.

In the right panel of fig. 3 we show $\sigma_0$ for $\sqrt{s} = 13$ TeV and $\sqrt{s} = 14$ TeV in the center of mass energy of the proton-proton system. The gain for 14 TeV compared to 13 ranges from 13% to 18% for $m_\eta = 63$ GeV and 150 GeV respectively. This computation is explained in the following section.

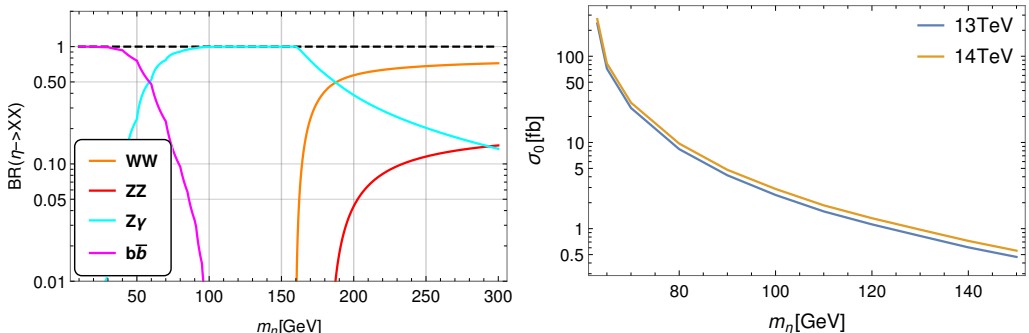

Figure 3: Left: Branching ratio of $\eta$ in the dominant decay channels, completely dictated by the anomalous interactions (proportional to $\kappa$) in eq. (3). The $b\bar{b}$ decay raises via loops of gauge bosons and is also completely fixed [32, 33]. Right: Cross section to the production process (1) with $\lambda = 1$, $\sigma_0$. Both $\sqrt{s} = 13$ TeV and 14 TeV in the $pp$ center-of-mass are depicted. More details on the computation are described in sec. 2.1.

## 2.1 Simulation setup

In order to understand the outcome of signal and SM background processes at the LHC, we performed simulations of $pp$ scattering using the MG5_AMC@NLO program [34] with MG5_AMC@NLO default dynamical factorization and renormalization scales, and NNPDF 2.3 LO parton distribution function (PDF) set with $\alpha_s(\mu) = 0.119$ [35]. Parton level events were processed through PYTHIA8 [36] for showering and hadronization and through DELPHES3 [37] for fast detector simulation. We used the default PYTHIA8 and CMS DELPHES cards.

To simulate the signal events and the total cross section $\sigma_0$ we used a UNIVERSAL FEYN-RULES OUTPUT (UFO) [38] model implemented locally. The model includes the SM tree-level interactions, the full energy dependence of the top-quark (and bottom-quark) triangle that goes in the interaction $g - g - h$ of diagram 1 and the interactions in the Lagrangian (3). [†]

We generated signal samples for all decay channels in which at least one of the $\eta$ particles decays into muons or electrons ($f\bar{f} = \ell^+\ell^-$, with $\ell = \mu, e$), while the other branch is split in 5 different channels with $f'\bar{f}' = \ell^+\ell^-$, $\tau^+\tau^-$, $jj$, $\nu\bar{\nu}$ and $b\bar{b}$, where jets $j$ are any light flavor quarks. We did not apply any kinematic cuts at parton level to the signal samples.

The simulation of the background is carried out analogously. In tab. 1 we show the total cross section for the relevant background processes

$$pp \rightarrow X + n_\gamma \gamma + \text{jets}, \tag{6}$$

where $X = \ell^+\ell^-\ell'^+\ell'^-$, $\ell^+\ell^-\ell'^\pm\nu$, $\ell^+\ell^-$ (or $4\ell$, $3\ell$ and $2\ell$ for short) and $n_\gamma$ is the number of *matrix element* (ME) photons. For the cases $4\ell$ and $3\ell$ we fix the maximum number of extra partonic jets (light quarks and gluons) such that the sum of jets plus ME photons is less or equal than 2. For example, for the 1 ME photon sample we sum a zero jet sample and a one jet sample. For the $2\ell$ sample instead we merge always up to two partonic jets no matter the photon multiplicity. The different jet multiplicities are merged with the MLM method [39]. To avoid double counting of hard photons, a matching condition has been implemented for photons as well, as we will soon discuss. The kinematic cuts shown in tab. 2 were applied to avoid divergences in the matrix elements and to avoid loss of statistics due to production of too many events outside the detector coverage. We have also considered the processes in (6) with

---

[†]Moreover, to discuss other production modes in sec. 4, we include the interactions (12), (16) and the coupling $g - g - \eta - \eta$ with the full triangle form factor of diagram 11. The UFO model is available upon request.

Table 1: Cross sections (fb) of background processes (6) for $n_\gamma = 0, 1, 2$ and $X = 4\ell, 3\ell, 2\ell$ in the format $\sigma_{LO}(K)$. $K_{14}$ is the correction factor to go from 13 to 14 TeV. Generation level cuts of tab. 2 are applied.

| $X$ \ $n_\gamma\gamma$ | $2\gamma$ | $1\gamma$ | $0\gamma$ | $K_{14}$ |
|---|---|---|---|---|
| $\ell^+\ell^-\ell'^+\ell'^-$ (4$\ell$) | $1.17 \times 10^{-2}(1.36)$ | $1.09 \times 10^{0}(1.34)$ | $5.53 \times 10^{1}(1.29)$ | 1.10 |
| $\ell^+\ell^-\ell'^\pm\nu$ (3$\ell$) | $1.17 \times 10^{-2}(2.88)$ | $7.94 \times 10^{0}(2.24)$ | $5.08 \times 10^{2}(1.62)$ | 1.09 |
| $\ell^+\ell^-$ (2$\ell$) | $1.27 \times 10^{-1}(1.50)$ | $2.71 \times 10^{1}(1.46)$ | $1.67 \times 10^{3}(1.27)$ | 1.08 |

Table 2: Parton level cuts performed for the generation of all the background samples, when they are applicable.

| $p_T(j) > 20$ GeV | $\Delta R(j, \gamma) > 0.4$ | $|\eta(\gamma)| < 2.5$ |
|---|---|---|
| $p_T(\ell) > 10$ GeV | $\Delta R(\ell^+, \ell^-) > 0.4$ | |
| $p_T(\gamma) > 10$ GeV | $\Delta R(\ell, \gamma) > 0.4$ | |

$X = \tau^+\tau^-, \ell^+\ell^-\tau^\pm\nu_\tau, \ell^+\ell^-\nu\bar\nu, t_{lep}\bar t_{lep}$, with $t_{lep}$ a top-quark decaying into a bottom-quark and leptons. These are all subdominant after the selections we discuss in sec. 3.

We applied a flat K-factor for each sample taken from the central value obtained from a next-to-leading order in QCD correction from Ref. [40]. For the 4$\ell$ samples we use the K-factor of $ZZ+0,1,2\gamma$ ($W^+W^-$ contribution is suppressed after selections described in sec. 3). For the 3$\ell$ and 2$\ell$ samples we use the K-factor from $WZ+0,1,2\gamma$ and $Z+0,1,2\gamma$ respectively. The K-factor for each background sample is displayed in the format $\sigma_{NLO} = \sigma_{LO}(K)$ in tab. 1. For the signal we applied a Higgs production NLO K-factor $K = 2.05$ also taken from [40].

The numbers in tab. 1 were obtained for a $pp$ center of mass energy of $\sqrt{s} = 13$ TeV. To estimate the event yields at $\sqrt{s} = 14$ TeV for HL-LHC we computed the total cross section of the base process (without extra photons or jets) using the same set of tools and applied a correction factor, $\sigma_{14} = \sigma_{NLO}K_{14}$. We checked that the difference in total cross section with the addition of extra photons or jets is negligible within the precision required for our analysis. For the analysis in sec. 3 we ignored further kinematic differences between 13 and 14 TeV, which is well justified by the inclusive character of our study. The correction factors $K_{14}$ to go from 13 to 14 TeV are reported in the last column of tab. 1.

## 2.2 Fake photons and matching

Photons identified in the calorimeters might have a different origin than the ME photons from the hard scattering. In our framework this identification is simulated by the fast simulation program DELPHES. The nature of the *reconstructed photons* provided by the DELPHES simulation can be obtained by looking at the particle at *truth* level (from PYTHIA8 ) originating it. If the reconstructed photon is isolated[‡], has $p_T > 10$ GeV and is radiated from a parton we label it as a *matched photon*.

All the events with a number of matched photons larger than the number of ME photons of the sample are discarded, since they are included in the sample with higher photon multiplicity. This matching procedure removes double counting. It does not apply to the sample with 2 ME photons because we did not generate samples with 3 or more ME photons, which are described by the shower MC program. In other words, in the sample with 0$\gamma$, (i.e. 0 ME photons) events

---

[‡]The isolation index $I$ is given by the scalar sum of $p_T$ of particles within a cone of $\Delta R = 0.4$ around the photon. The criterion for isolation is $I < 0.12$.

with at least one matched photon are discarded since they are accounted for by the sample with $1\gamma$, in the sample with $1\gamma$ all events with 2 or more matched photons are discarded, and for the $2\gamma$ sample no event is discarded. A similar algorithm has been implemented in the measurement of $t\bar{t} + \gamma$ [41].

After the matching procedure we identify 3 types of *fake photons*:

- Misidentified electron: Some electrons are missed in the tracker and leave only an energy deposit in the electromagnetic calorimeter (ECAL), which is hard to distinguish from a photon.

- Multi-particle origin: Some reconstructed photons are originated from more than one particle hitting the calorimeter. This type of photon comes typically from an electron and a photon (from radiation) or from 2 photons from electron conversion. The photons of this type are not matched because they are typically close to the electron.

- Photons from hadronic activities: These photons come from meson decays, mostly from $\pi^0 \to \gamma\gamma$. Experiments might be able to further reduce this background using information not contained in the DELPHES simulation.

We will use this classification to assess the impact of each type of fake photon as well as of the matching procedure in the event selection, to be described in the next section.

## 3 Analysis

The general strategy to search for the process in eq. (2) is to apply simple event selection using the standard reconstructed objects provided by DELPHES §. The strategy to choose the selection cuts is to optimize the significance (see eq. (9)) for the HL-LHC (3 ab$^{-1}$, $\sqrt{s} = 14$ TeV). We comment nevertheless on the sensitivity at Run III (300 fb$^{-1}$, $\sqrt{s} = 13$ TeV).

We start aiming at a clean reconstruction of one of the narrow $\eta$ resonances, thus demanding it to decay leptonically, i.e. $f\bar{f}$ in eq. (2) is a pair of same flavor opposite sign (SFOS) leptons (muons or electrons), which we also denote as $\ell^+\ell^-$. We require them to be separated by $\Delta R(\ell^+, \ell^-) > 0.4$. This selection removes background from collimated taus and $2\ell$ backgrounds. Moreover, we require at least two isolated photons. From the possible combinations of one SFOS and one photon, we reconstruct $\eta$ candidates and require the invariant mass of the system to be near a nominal $\eta$ mass within a 2 GeV mass window. These basic selections are summarized as,

$$\geq 1\ \text{SFOS}, \quad \Delta R(\ell^+\ell^-) > 0.4, \quad \geq 2\ \text{photons}, \quad |m(\ell^+\ell^-\gamma) - m_\eta| \leq 2\ \text{GeV}. \tag{7}$$

### 3.1 Leptonic channel

After the selection cuts eq. (7) the event yields are dominated by the $2\ell$ background (tab. 1), which can be drastically reduced via the requirement of a third lepton,

$$\geq 3\ \text{leptons}. \tag{8}$$

Therefore in this section we concentrate on the fully leptonic decay signal, where also the second $Z$ decays leptonically, or, better, $\eta \to \ell'^-\ell'^+\gamma$. In sec. 3.2 we discuss other strategies related to the semi-hadronic and semi-invisible decays (with one branch always decaying leptonically) which are not as powerful.

---

§The reconstructed photons have $p_T(\gamma) > 10$ GeV and $|\eta(\gamma)| < 2.5$, electrons (muons) have $p_T(\ell) > 10$ GeV and $|\eta(\gamma)| < 2.5(2.4)$. Apart from these basic features, different efficiency tables, isolation criteria and other features and objects are defined via DELPHES version 3.4.1 and the corresponding CMS card.

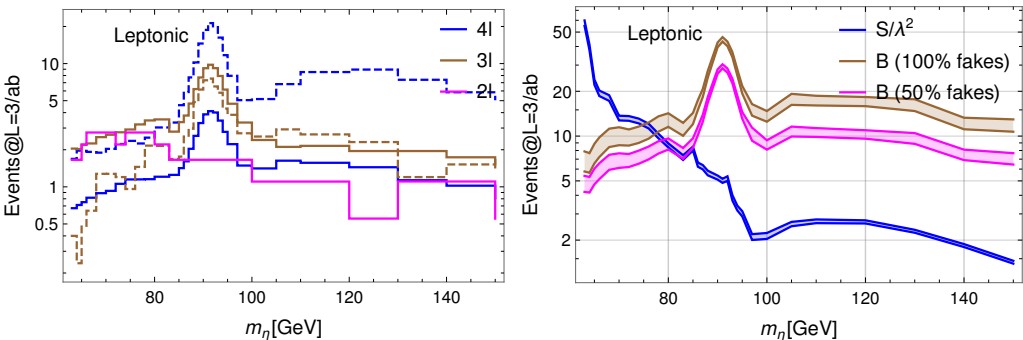

Figure 4: *Left:* Number of events for HL-LHC for the dominant background processes (6) after selections (7) and (8). The solid lines refer to samples with two ME photons ($n_\gamma = 2$) and the dashed lines to 1 ME photon ($n_\gamma = 1$). The dominant background for low $m_\eta$ are $3\ell 2\gamma$ and $2\ell 2\gamma$ and for high values of $m_\eta$ is $4\ell 1\gamma$ with a fake photon typically originating from an electron. *Right:* Total background (B) and signal (S) rates after selection cuts. The magenta curve is obtained with the reduction of 50% in fake photon rates. The bands indicates MC statistical error.

The total number of events for the dominant background processes is shown in the left panel of fig. 4. It is given by the cross section in eq. (4) times the efficiency of the selection (7)-(8). The solid curves stem from samples with 2 ME photons and the dashed ones for 1 ME photon (where the second selected photon is a fake photon). The blue curves refer to $4\ell$, the brown ones to $3\ell$, and the magenta to $2\ell$ backgrounds.

The dominant background for $m_\eta \gtrsim 85$ GeV is $4\ell+1\gamma$ (dashed blue) with a selected fake photon originating mainly from the forth electron. For low masses $m_\eta \lesssim 85$ GeV the $3\ell+2\gamma$ (solid brown) dominates, partly explained by its large QCD K-factor $K = 2.88$. The contribution from fake photons is suppressed due to the fact that there is not a forth electron to be misidentified. At low masses, a non-negligible contribution from $2\ell+2\gamma$ (solid magenta) is also present, with a fake lepton from hadronic activity or splitting of the photon into electrons. Due to the extremely low efficiency for this process (fake photon and lepton) we face a problem of MC statistics. To estimate this process yields we consider a larger mass window cut in $m(\ell^+\ell^-\gamma) = m_\eta \pm 8$ GeV and divide the result by 4. We take into account this MC error in our estimate of exclusion bounds. We estimate that the combination of a fake photon and a fake lepton drastically suppresses the $2\ell+1\gamma$ to be much lower than the $2\ell+2\gamma$. Other background processes are subdominant.

The total number of events for background (B) and signal (S) is shown in the right panel of fig. 4. For the signal we sum all possible decays, with yields dominated by the fully-leptonic channel and with an approximate 10% contribution from the $\ell\ell\tau\tau$ channel. The displayed numbers assume a coupling $\lambda = 1$ and scale like $\lambda^2$.

It is interesting to notice a drop in efficiency when the $Z$ is kinematically allowed to go on-shell, $m_\eta \gtrsim m_Z$, due to the fact that the available energy in the system is fully used by the $Z$ and the photon is extremely soft and unobserved. Once the available energy increases to $m_\eta \gtrsim m_Z + 10$ GeV, the photon is able to get some momentum and efficiency is recovered. The presence of light objects produced nearly at rest in the signal, combined with its low cross section, demands a low $p_T$ trigger for both photons and leptons. We used DELPHES recommendations: $p_T > 10$ GeV.

Fake photons might be further removed using detector information that is out of our simulation possibilities. In fig. 4 and in the following figures we also display the predictions for a background where the fake photon rate is reduced by 50% to illustrate how much a successful

Table 3: Probability (%) of having exactly one (=1) and at least two ($\geq 2$) matched and fake photons of each type to each background sample (6) and ME photon multiplicity $n_\gamma$. The numbers are extracted after the selection of $\geq 1$ SFOS and $\geq 3$ leptons. We did not include the $2\ell\,0\gamma$ background since this would require two fake photons and one fake lepton to pass the cuts. The * means that the events in this class are removed by the matching procedure due to double counting.

|  | ME ($n_\gamma$) | reconstructed | matched | electron | multi-part. | hadronic |
|---|---|---|---|---|---|---|
| $4\ell$ | 0 | =1 | 4.04* | 4.92 | 0.687 | 0.365 |
|  |  | $\geq 2$ | 0.0525* | $2.98\times10^{-4}$ | $\sim 0$ | $5.96\times10^{-4}$ |
|  | 1 | =1 | 65.6 | 5.01 | 0.768 | 0.340 |
|  |  | $\geq 2$ | 2.99* | $9.60\times10^{-3}$ | $\sim 0$ | $1.01\times10^{-3}$ |
|  | 2 | =1 | 39.1 | 4.98 | 0.684 | 0.367 |
|  |  | $\geq 2$ | 47.4 | 0.0171 | $\sim 0$ | $\sim 0$ |
| $3\ell$ | 0 | =1 | 1.38* | $7.77\times10^{-3}$ | 0.0146 | 0.343 |
|  |  | $\geq 2$ | 0.0117* | $4.86\times10^{-4}$ | $\sim 0$ | $\sim 0$ |
|  | 1 | =1 | 66.9 | 0.191 | 0.0854 | 0.322 |
|  |  | $\geq 2$ | 0.851* | $1.34\times10^{-3}$ | $\sim 0$ | $1.34\times10^{-3}$ |
|  | 2 | =1 | 39.0 | 0.269 | 0.140 | 0.313 |
|  |  | $\geq 2$ | 47.3 | 0.0108 | $\sim 0$ | $\sim 0$ |
| $2\ell$ | 1 | =1 | 2.04 | 0.571 | 0.214 | 0.286 |
|  |  | $\geq 2$ | $\sim 0$* | $\sim 0$ | $\sim 0$ | $\sim 0$ |
|  | 2 | =1 | 66.2 | 0.552 | $\sim 0$ | 0.276 |
|  |  | $\geq 2$ | 1.44 | $\sim 0$ | $\sim 0$ | $\sim 0$ |

implementation of such reduction by the experiment would affect the results.

These fake photons can have different origins, as discussed in sec. 2.2: electron, multi-particle and hadronic. The probability (%) of having exactly one (=1) or at least 2 ($\geq 2$) of each type of fake photon is shown in tab. 3. We display these numbers for the background processes (6). The numbers are extracted after the selection of $\geq 1$ SFOS and $\geq 3$ leptons. The corresponding numbers for matched photons are also shown.

The estimates for 2 fake photons of each type suffers from large statistical error, but they are indicative of their smallness.

For the $4\ell$ samples there is a large contribution $\sim 5\%$ from 1 fake electron, generated by the hard lepton not tagged as lepton. This is the reason for the dominance of the $n_\gamma = 1$ sample over the $n_\gamma = 2$ one due to the low cross section of the latter. We note that due to low cross section of the signal we cannot afford tagging an extra forth lepton to further suppress this background.

In the $3\ell$ samples there is no extra lepton to be misidentified, which reduces the fake electron rate to the permil level. Therefore, for this process the dominant fake contribution comes from hadronic activity. This fact makes the fake photon contribution subdominant w.r.t. the ME photon $n_\gamma = 2$ background.

The $2\ell$ samples have the further peculiarity of the presence of a fake lepton. The very low value of 2 matched photons in the 2 ME photon sample indicates that the 3rd selected lepton comes actually from a photon and thus for the final selection of 2 photons an extra fake photon is typically required even in the 2 ME photon sample.

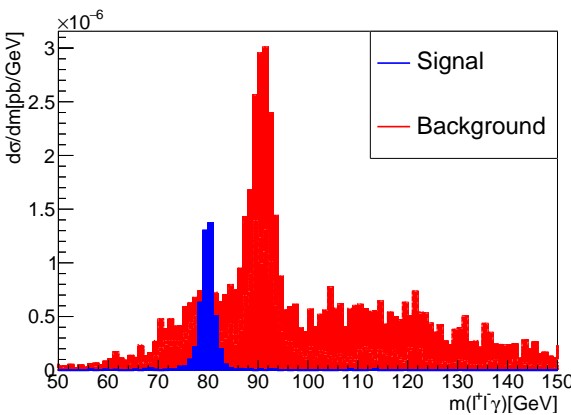

Figure 5: Invariant mass distribution of the best $\eta$ candidate for the $m_\eta = 80$ GeV hypothesis. The best candidate is defined as the $\ell^+\ell^-\gamma$ system with invariant mass nearest to $m_\eta$. The signal, in blue, presents a narrow peak at $m_\eta$ that can be used to discriminate from the background, in red.

It is also important to notice that the matching procedure plays an important role in our estimates, reducing the background with less than 2 ME photons. The reduction in total cross section is small, typically of the order of %. However, the photons from radiation of a $Z$ decay, e.g. $Z \to e^+e^-\gamma$ tend to mimic better the photons from the signal. These photons are removed if generated by the shower program to avoid double counting (under the matching conditions discussed in sec. 2.2) and thus, after selection cuts in eqs. (7)–(8), the overall reduction can reach approximately 90%.

Besides counting photons and leptons, the main discriminating observable is provided by the mass of the best $\eta$ candidate ($\ell^+\ell^-\gamma$-system with invariant mass closest to $m_\eta$), which presents a sharp peak at $m_\eta$. This distribution is shown in fig. 5 after the cuts in eqs. (7)–(8) (removing the $\eta$ mass window cut). The signal hypothesis is for $m_\eta = 80$ GeV.

After estimating signal ($S$) and background ($B$) yields, we compute the significance with the formula

$$z = \sqrt{2}\sqrt{(B+S)\log\left(\frac{(B^2\Delta^2+B)(B+S)}{B^2\Delta^2(B+S)+B^2}\right) - \frac{\log\left(\frac{\Delta^2 S}{B\Delta^2+1}+1\right)}{\Delta^2}}. \tag{9}$$

We denote by $\Delta = \sigma_B/B$ the percentage systematic error. Formula 9 allows one to take the relative systematic error $\Delta$ into account, extending the well known formula for the significance $z = \sqrt{2}\sqrt{(S+B)\log\left(\frac{S+B}{B}\right) - S}$. Indeed, it reduces to it in the limit $\Delta \to 0$. Both formulas are obtained using the Asimov data-set [42] into the profile likelihood ratio [43, 44] and is explicitly written in Ref. [45]. In the following we assume a systematic uncertainty of $\Delta = 10\%$. The uncertainty for this search is strongly dominated by statistics and varying $\Delta$ has a mild effect on our results.

The expected upper bound on $\lambda$ at 95% of confidence level (CL) (we solve eq. (9) for $z = 2$, corresponding to $\approx 95.45\%$ CL) is shown in fig. 6 for HL-LHC (left) and for Run III (right).

## 3.2 Hadronic and invisible channels

One of the main difficulties of this analysis is the low signal cross-section. This is not only due to the small cross section for double $\eta$ production but also to the small branching ratio of $Z \to \ell^+\ell^-$. It is therefore interesting to consider additional decay channels in which one of

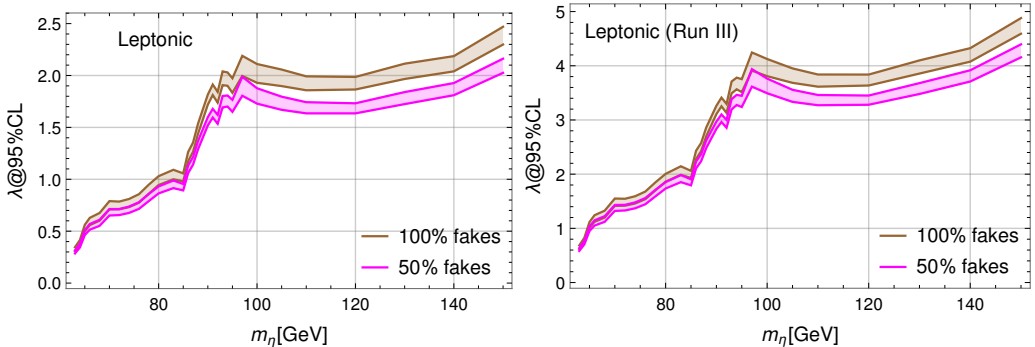

Figure 6: Projected upper bound on $\lambda$ at 95%CL at HL-LHC (left) and Run III (right). The magenta curve is obtained with the reduction of 50% in fake photon rates. The bands indicate MC statistical error.

the two $Z$ is allowed to decay hadronically or invisibly. The outcome is that these channels do not give competitive bounds w.r.t. the fully-leptonic channel, but we nevertheless report the results here for completeness and eventual future improvements.

For both channels we apply the same set of basic cuts in eq. (7), i.e. we want to fully reconstruct one $\eta$ via its leptonic decay as well as requiring at least two photons. Since only one $Z$ decays leptonically now, we do not require a third lepton, eq. (8), anymore, but instead require the presence of a system composed of a photon (one of those not identified as part of the best leptonic $\eta$ candidate) and one of the two options:

- Two jets for the hadronic selection, relevant to $Z \to q\bar{q}$.

- Vectorial missing transverse energy $\mathbf{E}_{\mathrm{T}}^{\mathrm{miss}}$ for the invisible selection relevant to $Z \to \nu\bar{\nu}$.

For the hadronic selection, the jets are clustered using the anti-$k_T$ algorithm with $p_T > 20$ GeV. We further demand the invariant mass of the $jj\gamma$ system to be

$$m(jj\gamma) > m_\eta - 20 \text{ GeV}. \tag{10}$$

For the invisible selection, we require the transverse momentum of the $\mathbf{E}_{\mathrm{T}}^{\mathrm{miss}}\gamma$ system to be

$$p_T(\mathbf{E}_{\mathrm{T}}^{\mathrm{miss}}\gamma) > 40 \text{ GeV}. \tag{11}$$

The resulting number of events and exclusion limit on $\lambda$ for HL-LHC are shown in fig. 7 for the hadronic selection and fig. 8 for the invisible selection. We use the same color and style conventions of fig. 6 and fig. 4.

We can see that these two channels are never competitive with the fully leptonic one, at least if performing the basic counting analysis described above.

## 4 Composite Higgs models and other production modes

As a concrete example of a model presenting the features that motivate our study we consider a non linearly realized Higgs sector based on the global symmetry breaking $SO(6)/SO(5) = SU(4)/Sp(4)$, comprising the usual Higgs doublet plus an EW singlet $\eta$ [46]. In the spirit of partial compositeness [7], this model can also be augmented with top partners coupling linearly to the third generation quark fields $Q_L = \begin{pmatrix} t_L \\ b_L \end{pmatrix}$ and $t_R$. The underlying

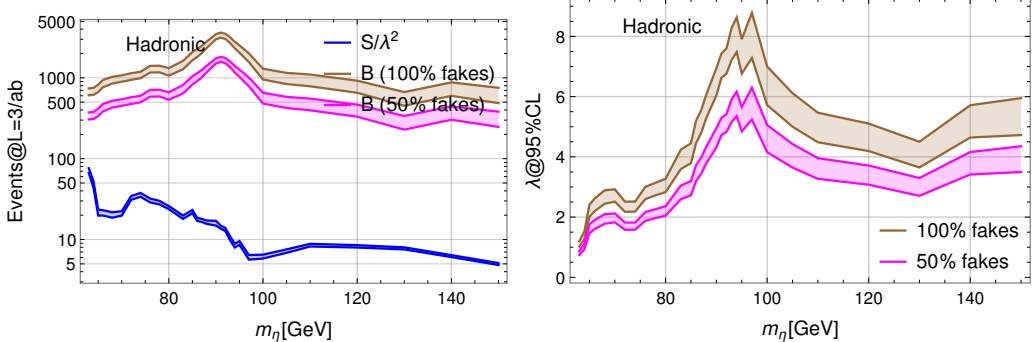

Figure 7: Hadronic selection. *Left:* Number of events for HL-LHC after hadronic selection (7) and (10) for signal (blue) and background, with all fake photons (brown), and assuming a 50% reduction in fake photons (magenta). *Right:* 95%CL upper bound on $\lambda$ from hadronic selection.

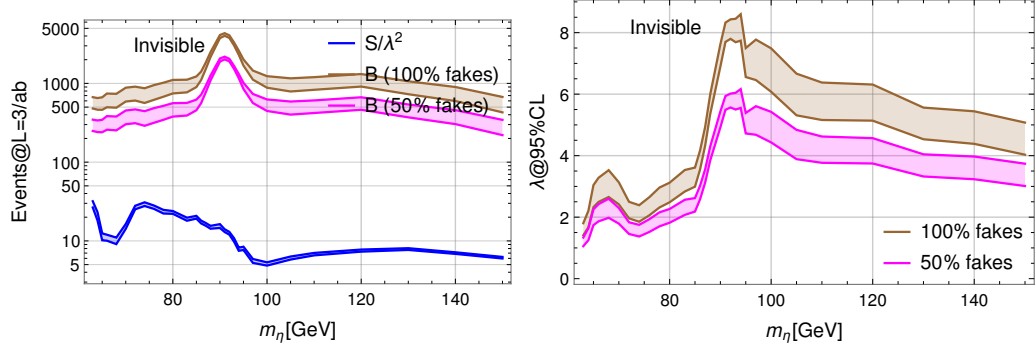

Figure 8: Invisible selection. *Left:* Number of events for HL-LHC after invisible selection (7) and (11) for signal (blue) and background, with all fake photons (brown), and assuming a 50% reduction in fake photons (magenta). *Right:* 95%CL upper bound on $\lambda$ from invisible selection.

gauge theory introduces new hyperfermions charged under a new confining hypercolor group $Sp(4)$ [12]. The hyperfermions combine into hypercolor singlet trilinears top partners providing useful guidance on the possible nature of the spurion embedding, i.e. under what kind of (incomplete) irreducible representation (irrep) of $SU(4)$ the fields $Q_L$ and $t_R$ transform. We consider different possibilities and show that some cases have an $\eta$ with the required properties: its linear couplings to SM fermions, including the top quark, are suppressed, and its mass can fit in the range $m_h/2 < m_\eta \lesssim 150$ GeV. We also discuss other production mechanisms that might compete with the off-shell Higgs process here studied. For details on the conventions we refer the reader to [15].

## 4.1 EW gauge interactions and vector boson fusion

The $\eta$ couplings to weak bosons are particularly rigid, driven by the leading dimension kinetic operator of the chiral Lagrangian,

$$\mathcal{L} \supset \frac{f^2}{8} D_\mu U D^\mu U^\dagger \supset \left( M_W^2 W^{+,\mu} W_\mu^- + \frac{M_Z^2}{2} Z^\mu Z_\mu \right) \left( 1 + \frac{2\cos\theta}{v} h - \frac{\sin^2\theta}{v^2} \eta^2 \right), \qquad (12)$$

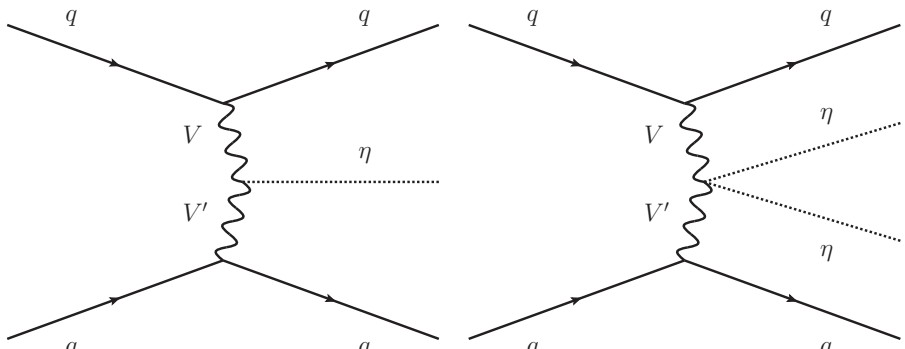

Figure 9: VBF Feynman diagrams for single production (left) and pair production (right).

that also defines the misalignment angle $v = f \sin\theta$, where $M_W^2 = \frac{g^2 v^2}{4}$ and $M_Z^2 = \frac{(g^2+g'^2)v^2}{4}$, and $f \gtrsim 800$ GeV the pNGB decay constant. For the description of the scalar fields as pNGB we use a $4 \times 4$ unitary matrix $U$ transforming as $U \to g U g^T$ under $SU(4)$. Moreover, linear couplings in $\eta$ are generated by the WZW anomaly parametrized by the term proportional to $\kappa$ in eq. (3). The $\kappa$ coefficient is given by

$$\kappa = 2\sin\theta\cos\theta, \tag{13}$$

for the hyperfermion $\psi$ transforming in the fundamental representation of the hypercolor group $Sp(4)$. Any sensible value of $\kappa$ forces a very narrow $\eta$, with a total width $\Gamma \lesssim \mathcal{O}(10)$ keV.

These interactions fix the production rates via VBF, either single production via the anomaly in eq. (3) or double production via eq. (12), as depicted in the VBF diagrams in fig. 9 on the left and right respectively. The cross section of this type of production in proton collision is small w.r.t. the off-shell Higgs production, as shown in fig. 10. The estimate was obtained using the simulation setup described in sec. 2 with $\sqrt{s} = 14$ TeV in the proton-proton center of mass and a generation cut on the jets' transverse momenta $p_T(j) > 10$ GeV and $\Delta R(jj) > 0.4$ between the jets. For the off-shell Higgs production we used $\lambda = \kappa_t \lambda_\eta = \cos^2\theta$ (see tab. 4 and following discussion for more details).

We notice that these interactions are common to other model realizations, in particular in PC based on the coset $SU(4) \times SU(4)/SU(4)$ [15, 31]. They do not depend strongly on the mechanism to give mass to fermions, for instance via a bilinear condensation [47].

## 4.2 $\eta$-fermion interaction and its contribution to pair production

We now analyze the additional features arising when introducing top partners in the model. Since the allowed top partners of [12] may transform under the **1** (singlet), **6** (antisymmetric) or **15** (adjoint) of $SU(4)$, in order to allow for the simplest linear coupling between them and the SM quarks we chose the SM quarks to be embedded in those same irreps. (Obviously the singlet is a viable choice only for $t_R$.) These irreps also allow for embeddings of $Q_L$ satisfying the requirements imposed by the $Z \to b\bar{b}$ constraints [51]. Note that they are all real irreps of $SU(4)$.

Keeping in mind the reduction of these irreps into irreps of the custodial $SU(4) \to SU(2)_L \times SU(2)_R$

$$\mathbf{1} \to (\mathbf{1},\mathbf{1}), \quad \mathbf{6} \to 2 \times (\mathbf{1},\mathbf{1}) + (\mathbf{2},\mathbf{2}), \quad \mathbf{15} \to (\mathbf{1},\mathbf{1}) + 2 \times (\mathbf{2},\mathbf{2}) + (\mathbf{3},\mathbf{1}) + (\mathbf{1},\mathbf{3}), \tag{14}$$

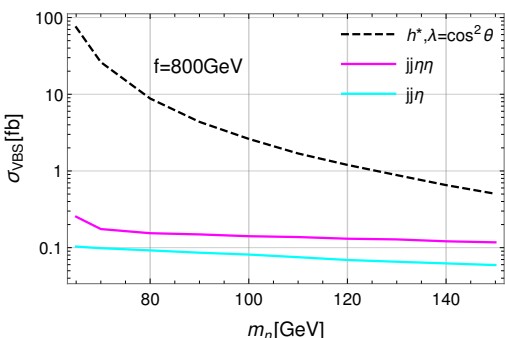

Figure 10: Production cross section, for $\sqrt{s} = 14$ TeV, via VBF, either $\eta$ singly produced through anomalous interactions (cyan) or doubly produced (magenta) and via off-shell Higgs mechanism (black dashed), with $\lambda = \kappa_t \lambda_\eta = \cos^2\theta$, for comparison.

we see that we can embed $Q_L$ in a unique way into **6** and in two ways into **15**. Similarly, $t_R$ can be embedded in one way in **1**, two ways into **6** and four ways into **15** ($(\mathbf{1}, \mathbf{1})$ and $(\mathbf{1}, \mathbf{3})$).

We denote the explicit embedding as $Q_{L\mathbf{n}} = t_L S^{\mathbf{n}}_{t_L} + b_L S^{\mathbf{n}}_{b_L} \equiv q_L^a S^{\mathbf{n}}_{q_L^a}$ and $t^c_{R\mathbf{n}} = t^c_R S^{\mathbf{n}}_{t_R}$, where $\mathbf{n} = \mathbf{1}, \mathbf{6}$ or $\mathbf{15}$ and $S$ are the generic numerical spurionic matrices, normalized as $\text{tr}(S.S^\dagger) = 1$. We use left-handed fields throughout, hence the charge conjugation operation $^c$ on $t_R$.

In the case of multiple possible embeddings we use an angular variable to parameterize the choice. For instance $S^{\mathbf{6}}_{t_R} = \sin\alpha_R S^{I\mathbf{6}}_{t_R} + \cos\alpha_R S^{II\mathbf{6}}_{t_R}$, where $S^{I\mathbf{6}}_{t_R}$ and $S^{II\mathbf{6}}_{t_R}$ are the singlets of $SU(2)_L \times SU(2)_R$ and $Sp(4)$ respectively. In the same way $S^{\mathbf{15}}_{q_L^a} = \sin\alpha_L S^{I\mathbf{15}}_{q_L^a} + \cos\alpha_L S^{II\mathbf{15}}_{q_L^a}$. The explicit expression for $S^{\mathbf{15}}_{t_R}$ is not needed in what follows, since only one irrep works.

Our first task is to find how the different choices of spurions generate the top quark mass while at the same time forbidding the presence of a $\eta\bar{t}t$ coupling. Writing the contribution to the Lagrangian as $\mathcal{L} \supset y_Q y_t f \mathcal{O} + \text{h.c.}$, where $y_Q$ and $y_t$ are the pre-Yukawa couplings, a systematic analysis shows that the following operators meet the above minimal requirements:

$$
\begin{aligned}
\mathcal{O}_{\mathbf{6},\mathbf{1}} &= \text{tr}(Q_{L\mathbf{6}} U^*) t^c_{R\mathbf{1}}, \\
\mathcal{O}_{\mathbf{6},\mathbf{15}} &= \text{tr}(Q_{L\mathbf{6}} U^* t^c_{R\mathbf{15}}) \;\; (\text{with } T^3_R = 0), \\
\mathcal{O}_{\mathbf{6},\mathbf{6}} &= c\, \text{tr}(Q_{L\mathbf{6}} U^*) \text{tr}(t^c_{R\mathbf{6}} U^*) + c'\, \text{tr}(Q_{L\mathbf{6}} U^* t^c_{R\mathbf{6}} U^*) \;\; (\text{with } \alpha_R = 0), \\
\mathcal{O}_{\mathbf{15},\mathbf{6}} &= \text{tr}(Q_{L\mathbf{15}} t^c_{R\mathbf{6}} U^*).
\end{aligned}
\tag{15}
$$

A few remarks are in order. For $\mathcal{O}_{\mathbf{6},\mathbf{15}}$, only the $T^3_R = 0$ component of $(\mathbf{1}, \mathbf{3})$ fulfills our requirements. For the case with both $Q_L$ and $t^c_R$ in the **6** there are two possible leading dimension operators (added in $\mathcal{O}_{\mathbf{6},\mathbf{6}}$ with arbitrary coefficients), but only with $\alpha_R = 0$, i.e. using the $Sp(4)$ singlet can we avoid a $\eta\bar{t}t$ coupling. (This is well known from [46].) On the other hand, in the case $\mathcal{O}_{\mathbf{15},\mathbf{6}}$ the absence of $\eta\bar{t}t$ coupling is generic. Note that the case where both $Q_L$ and $t^c_R$ are in the **15** does not yield any non-trivial leading dimension invariant given the necessity to multiply $U$ and $U^*$ directly.

Expanding the operators 15 we can read off the top quark mass and its coupling to the Higgs boson and the $\eta$,

$$
\mathcal{L} \supset -m_t \left( 1 + \frac{h}{v}\kappa_t - \frac{h^2}{f^2}\kappa_{th^2} - \frac{\eta^2}{f^2}\kappa_{t\eta^2} \right) \bar{t}t,
\tag{16}
$$

with coefficients $\kappa_t$, $\kappa_{th^2}$ and $\kappa_{t\eta^2}$ given in tab. 4. The two invariants in $\mathcal{O}_{\mathbf{6},\mathbf{6}}$ give the same contribution to $m_t$ and the couplings and can thus be added together. From a detailed analysis

Table 4: Couplings between $\eta$, Higgs and the top quark for the coset $SU(4)/Sp(4)$. The numbers in the first two columns refer to the dimensions of the $SU(4)$ spurion irreps. The couplings are defined in eq. (16) and eq. (3).

| $Q_L$ | $t_R$ | $\kappa_t$ | $\kappa_{th^2}$ | $\kappa_{t\eta^2}$ | $\lambda_\eta$ | comments |
|---|---|---|---|---|---|---|
| **6** | **1** | $\cos\theta$ | $1/2$ | $1/2$ | $\cos\theta$ | |
| **6** | **15** | $\cos\theta$ | $1/2$ | $1/2$ | $\cos\theta$ | $T_R^3 = 0$ of $(\mathbf{1},\mathbf{3})$ |
| **6** | **6** | $\cos(2\theta)/\cos\theta$ | $2$ | $1$ | $\cos\theta$ | $\alpha_R = 0$ |
| **15** | **6** | $\cos\theta$ | $1/2$ | $1/2$ | $\cos\theta$ | |

of the potential (see sec. 4.3) we also find that

$$\lambda_\eta = \cos\theta\,. \tag{17}$$

The $\eta^2 t\bar{t}$ contact interaction in eq. (16) allows a new type of contribution to $\eta$ pair production, depicted in diagram 11 ¶. The total cross section of pair production of $\eta$, including both diagrams, for $\sqrt{s} = 14$ TeV, $f = 800$ GeV and different values of $m_\eta$, is shown on the left plot of fig. 12. The solid lines refer to the two most promising top representations that can provide a low $\eta$ mass (see subsection below), $(Q_L, t_R^c) = $ **(15,6)** and **(6,6)**. The corresponding dashed lines instead depict the pure off-shell Higgs contribution. The lower panel shows the ratio between the full result and the pure off-shell Higgs.

In fig. 12 (left) a reduction in cross section for low mass can be noticed. This happens due to the destructive interference between diagrams 11 and 1. Eventually, either for large $m_\eta$ where the Higgs offshellness becomes prohibitive, or for large $\sin\theta$ (low compositeness $f$), the contact interaction dominates the production mechanism. In this sense, these two interactions have complementary role in excluding different regions of parameter space - while off-shell Higgs dominates for high value of $f$ and low $m_\eta$, the contact interaction dominates for low $f$ and large $m_\eta$. The combination of them allows to exclude a large part of $f$, shown on fig. 12 to the right. For that exclusion region we assumed the efficiencies of the leptonic selection cuts discussed in sec. 3 to be unmodified. We took the central value prediction for both signal and background.

The region of low $\sin\theta$, where the off-shell Higgs mechanism dominates and give sensitivity, is preferred by data. Higgs coupling measurements give a direct bound to all models, $f \gtrsim 460$ GeV ($\sin\theta \lesssim 0.53$) at 2 standard deviations [20]. Electroweak precision observables give model dependent constraints typically $f \gtrsim 1$ TeV ($\sin\theta \lesssim 0.23$) [54]. Lower values of $f$ are possible and natural if cancellations with the composite vectors are present $f \gtrsim 670$ GeV, and even lower if the scalar excitation mass is below TeV [55].

Thus, the mechanism of off-shell Higgs production discussed in the previous sections is the relevant one to exclude the region of most physical interest, i.e. the lower left corner of fig. 12 (right). The conclusion we reach is that the contact interaction $\eta^2 t\bar{t}$ is typically present in more complete models, but does not affect the off-shell Higgs sensitivity in the relevant light $\eta$ mass region. The relevance of the additional $h^2 t\bar{t}$ interaction in double Higgs production has been discussed in [56].

Let us also briefly comment on other realizations. In PC based on $SU(4) \times SU(4)/SU(4)$ with $Q_L$ in the adjoint and $t_R$ in the singlet of $SU(4)$, we find the same interactions 16 with $\kappa_t = \cos\theta$ and $\kappa_{t\eta^2} = 1/2$. If the top mass is generated by a bilinear operator (as in extended

---

¶The contact interaction $\eta^2 t\bar{t}$ in eq. (16) can also be regarded in models of PC as arising from integrating out heavy top partner states $T$ with off-diagonal couplings of type $t - T - \eta$ [52,53]. Due to the high masses of such states compared to the typical energy of the process studied in this work finite mass effects are expected to be negligible.

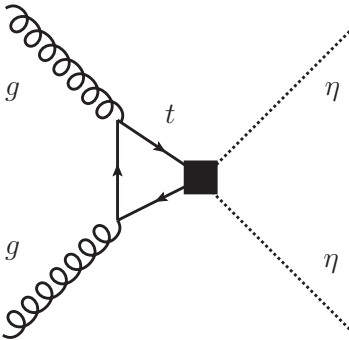

Figure 11: Extra diagram contributing to $\eta$-pair production.

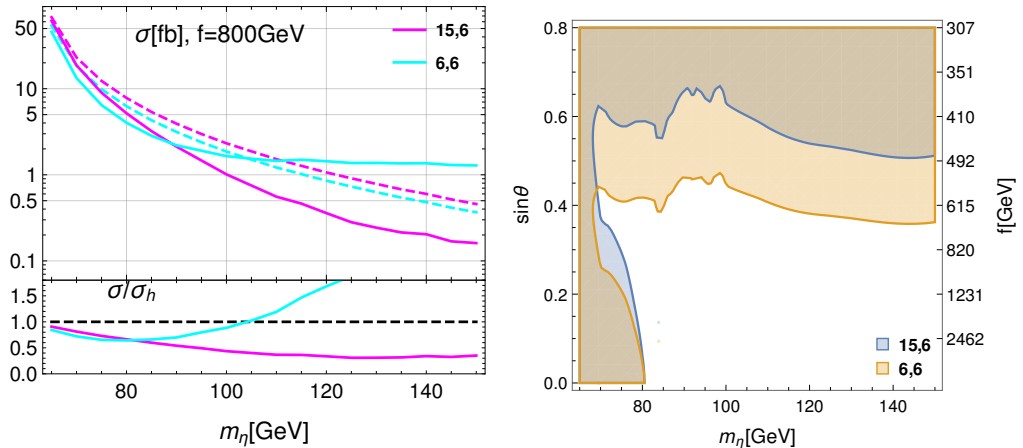

Figure 12: *Left:* Total $\eta$ pair production cross section at 14 TeV LHC for $Q_L$, $t_R^c$ in the **15, 6** (cyan) and **6, 6** (magenta) including the coherent sum of contact (fig. 11) and off-shell Higgs (fig. 1) contributions in solid lines, and only the off-shell Higgs in dashed lines. The lower panel show the ratio of the full calculation over the pure off-shell Higgs contribution. *Right:* Excluded region in $(m_\eta, \sin\theta)$ space for the same choice of spurions, obtained within the framework described in sec. 3 using the leptonic selection.

Technicolor theories) the coefficients are instead $\kappa_t = \cos\theta$ and $\kappa_{t\eta^2} = 1$ [47]. On the other hand, in models where the Higgs is a mixture of composite and elementary states and the condensate is not responsible for the fermion masses, the contact interaction is expected to be suppressed $\kappa_{t\eta^2} \ll 1$ [48–50].

## 4.3 $\eta$ mass and Higgs coupling $\lambda_\eta$

In this section we estimate the values of $m_\eta$ and $\lambda_\eta$ for the underlying gauge theory above. We do this by constructing the potential arising by explicitly breaking the global $SU(4)$ symmetry via spurion insertions. We work with only two spurion insertions. The full set of higher order terms has been computed in [57] for this and other models (see also [58]).

The scalar potential consists of the three following contributions. The first one is the contribution of the hyperquark masses

$$V_m = B_m f^4 \operatorname{tr}(\epsilon_0 U) + \text{h.c.} \tag{18}$$

We use the decay constant $f$ as the only dimensional parameter and denote the low energy coefficients (LEC) by dimensionless quantities such as $B_m$. We have taken the hyperquark mass proportional to $\epsilon_0$ as required if one wants to leave the full $Sp(4)$ unbroken. This is not strictly necessary, a more generic term preserving only the custodial group could be allowed, although we do not consider this case.

The second contribution comes from the SM EW gauge bosons

$$V_g = B_g f^4 \, \mathrm{tr} \left( g^2 T_L^A U T_L^{AT} U^\dagger + g'^2 T_R^3 U T_R^{3T} U^\dagger \right). \tag{19}$$

The sign of the LEC $B_g$ is known to be positive [59].

The third contribution, triggering vacuum misalignment [60], comes from the spurions for the quarks of the third family. It can be written, for the four choices of interest presented above, as

$$
\begin{aligned}
Q_{L6}, t_{R1}^c: \quad & V_t = -B_t f^4 y_Q^2 \mathrm{tr}(S_{q_L^a}^{\mathbf{6}} U^*) \mathrm{tr}(S_{q_L^a}^{\mathbf{6}} U^*)^* \\
Q_{L6}, t_{R15}^c: \quad & V_t = -B_t f^4 \left( y_Q^2 \mathrm{tr}(S_{q_L^a}^{\mathbf{6}} U^*) \mathrm{tr}(S_{q_L^a}^{\mathbf{6}} U^*)^* + y_t^2 \mathrm{tr}(S_{t_R}^{\mathbf{15}} U S_{t_R}^{\mathbf{15}*} U^*) \right) \\
Q_{L6}, t_{R6}^c: \quad & V_t = -B_t f^4 \left( y_Q^2 \mathrm{tr}(S_{q_L^a}^{\mathbf{6}} U^*) \mathrm{tr}(S_{q_L^a}^{\mathbf{6}} U^*)^* + y_t^2 \mathrm{tr}(S_{t_R}^{\mathbf{6}} U^*) \mathrm{tr}(S_{t_R}^{\mathbf{6}} U^*)^* \right)|_{\alpha_R=0} \\
Q_{L15}, t_{R6}^c: \quad & V_t = -B_t f^4 \left( y_Q^2 \mathrm{tr}(S_{q_L^a}^{\mathbf{15}} U S_{q_L^a}^{\mathbf{15}*} U^*) + y_t^2 \mathrm{tr}(S_{t_R}^{\mathbf{6}} U^*) \mathrm{tr}(S_{t_R}^{\mathbf{6}} U^*)^* \right),
\end{aligned}
\tag{20}
$$

where $B_t$ is the third and last dimensionless LEC and we sum over weak isospin $a = 1, 2$.

We can now put together the three contributions $V = V_m + V_g + V_f$ and analyze the ensuing spectrum and couplings. Some very generic relations arise, allowing us to pick the models that satisfy our requirements. For all possible choices of spurions we find that $\lambda_\eta = \cos\theta$, as already shown in tab. 4.

The scalar masses are also simply related to each other as

$$
\begin{aligned}
Q_{L6}, t_{R1}^c: \quad & \frac{m_\eta^2}{f^2} = \frac{m_h^2}{v^2} \\[4pt]
Q_{L6}, t_{R15}^c: \quad & \frac{m_\eta^2}{f^2} = \frac{m_h^2}{v^2} \\[4pt]
Q_{L6}, t_{R6}^c: \quad & \frac{m_\eta^2}{f^2} = \frac{m_h^2}{v^2} + 8 y_t^2 B_t \\[4pt]
Q_{L15}, t_{R6}^c: \quad & \frac{m_\eta^2}{f^2} = \frac{m_h^2}{v^2} + 8 y_t^2 B_t \cos 2\alpha_R.
\end{aligned}
\tag{21}
$$

The last two scenarios are the only ones allowing an $\eta$ lighter that $h$. These expressions also illustrate the fine tuning issue in this class of models. Since the origin of the potential is the same as the one generating the top mass, we expect the terms proportional to $8y_t^2 B_t$ in eq. (21) to be order 1, which then competes with the first term $\frac{m_h^2}{v^2}$ to provide the $\eta$ mass. However, if $f$ is very large a fine cancellation between these two terms is necessary.

## 5 Conclusions

The production of a pair of pseudoscalars $\eta$ through a Higgs propagator below the mass threshold ($m_h < 2m_\eta$) has received little attention. However, this mass region is just as motivated as others from the point of view of model building. Guided by constructions via underlying gauge theories, we considered such process with $\eta$ both fermiophobic and photophobic. The

$\eta$ decays almost exclusively into $Z\gamma$ in the mass region $\frac{m_h}{2} < m_\eta < 2m_W$ and is extremely narrow, being produced on-shell despite the presence of possible off-shell Higgs and $Z$ bosons. The main phenomenological result of our work is shown in fig. 6, where we provide a projection on the sensitivity of the signal strength $\lambda$ (eq. (4)) in such scenarios. With $\lambda < 1$ we can probe $\eta$ masses up to around 70 GeV for the Run III data-set, and around 85 GeV for the future HL-LHC. Beyond this mass, the on-shell SM $Z$ production becomes too large, requiring a substantial enhancement of the signal strength.

We performed a detailed analysis of the signal and the background, considering at least one leptonically decaying $Z$ boson (either on-shell or off-shell) and reconstructing the narrow $\eta$ state from the system $\ell^+\ell^-\gamma$. The most promising final state turns out to be the fully leptonic one ($\eta\eta \to \ell^+\ell^-\gamma\ell'^+\ell'^-\gamma$). This is so because despite the low signal rate the background can be highly suppressed by requiring $\geq 2$ photons and $\geq 3$ leptons. Other final states, including one of the $Z$ bosons decays hadronically or invisibly, have also been considered and give weaker bounds.

A good photon identification, as well as a reduction of fake photons, is also relevant for the search. Lacking the possibility to do a fully realistic simulation of the experimental apparatus, we simply show for comparison the results in the case of a 50% fake photon reduction. We employ a method to match different photon multiplicities between samples, which is important for the correct description of multi-photon processes.

To motivate the phenomenological analysis with a concrete model, in sec. 4 we considered a CH model based on an underlying gauge theory with PC mechanism to give mass to the top quark. We showed that several top partner representations predict a fermiophobic $\eta$, and two of them can give rise to a light $\eta$ state. The interpretation of our predicted bounds on $\lambda$ in terms of the model parameters $(m_\eta, f)$ is given in fig. 12. This result has an interesting implication for CH models, since it allows us to exclude large values of $f$ (for $m_\eta$ small), thus being complementary to other production mechanisms. We have also discussed the other possible production mechanisms: single and pair VBF production (fig. 9) and $\eta$ pair production via top loop (fig. 11). They are all sub-leading with respect to the off-shell Higgs production for values of $f$ of interest.

Pair production of $\eta$ via an off-shell Higgs is an experimentally challenging process which will require the full capabilities of the HL-LHC and will allow us to probe an interesting class of CH models.

# Acknowledgements

DBF and GF are supported by the Knut and Alice Wallenberg foundation under the grant KAW 2017.0100 (SHIFT project). DBF would like to thank Michele Selvaggi for answering questions regarding fake photons in Delphes. JS is supported by the National Natural Science Foundation of China (NSFC) under grant No.11947302, No.11690022, No.11851302, No.11675243 and No.11761141011 and also supported by the Strategic Priority Research Program of the Chinese Academy of Sciences under grant No.XDB21010200 and No.XDB23000000. LH is funded in part by United States Department of Energy grant number DE-SC0017988 and the University of Kansas Research GO program.

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
