# Peer review of "Pseudoscalar pair production via off-shell Higgs in composite Higgs models"

_SciPost Physics, doi:SciPost Phys. 9, 077 (2020)_

## Round 1 · Referee Report · Anonymous (Referee 1) · 2020-7-2

Strengths

1) Complete phenomenological study, from detailed model-building to collider 2) Impeccable LHC simulation and search proposal 3) Well motivated and novel

Weaknesses

1) Proposed signature, while possible, is non generic

Report

This wel-written article provides a thorough of composite-Higgs models in which the Higg sector contains an additional scalar singlet field (as Pseudo Goldstone boson). The authors show that in certain regions of parameter space (possible and protected by a symmetry) the extra singlet decays predominantly to Zgamma, and a favorable search channel is h_>eta eta->ZgammaZgamma. The article provides a detailed study of the discovery perspective at LHC (for 300 and 3000 1/fb).

Requested changes

I've a few minor comments: 1) In the intro (p2) it is said that the singlet "couplings to fermions are also typically small". It seems to me that they rather "can be small" in certain regions of parameter space protected by a symmetry (i.e. a generic embedding in the 6 representation would have sizeable fermion couplings) 2) In the statistical analysis (p11) the authors provide formula 3.3. This appears to me slightly out of context: I'd rather provide more details about how this formiula is used, or omit it. 3) In the conclusions (p19) it is said that "...allows us to exclude large values of f". This is not completely obvious from fig 12 (for which the bulk of the exclusion is at small f); so perhaps the authors could specify that they refer to the small m_eta region, if this is what is meant.

  • validity: top
  • significance: high
  • originality: high
  • clarity: top
  • formatting: perfect
  • grammar: perfect

Author:  Diogo Buarque Franzosi  on 2020-09-11  [id 957]

(in reply to Report 1 on 2020-07-02)
Category:
answer to question

Thank you for your comments and sorry for the late reply. We addressed your questions below and modified the paper as indicated.

1) We rephrased the sentence emphasizing that we are interested in scenarios where the coupling is small and referring to the original model building literature.

2) We provided more detail about the formula (3.3) just below it.

3) Yes, this is what we meant. We added the comment that we refer to small m_eta.

---

## Round 1 · Referee Report · Anonymous (Referee 2) · 2020-9-5

Report

The paper discusses the collider signatures of a light pseudo-scalar decaying to $Z\gamma$. The scenario is motivated by a class of composite Higgs models. A detailed analysis of the LHC reach is performed and the relevance of photon identification is stressed.

I think the paper presents new interesting results and is well-written. Before being able to recommend it for publication, however, I would ask the authors to make a few adjustments:

— In the Introduction: why is the coupling of $\eta$ to $\gamma\gamma$ small/absent? Please refer to an explicit model or postpone an explanation of how/why this happens to Section 4.

— Also, why isn’t there a CP-odd coupling to gluons? How much do your results depend on the absence of it?

— The sentence “.. via doubly suppressed anomalous interactions in a loop…” is rather unclear, especially if what the authors mean is essentially what is explained about the $\eta\to bb$ rate when discussion Fig 3.

— Still in the Introduction, in the paragraph below (1.1) it is said that “in models of PC this assertion needs to be better qualified”. Please anticipate that this is done in a later section, if this is the case.

— Regarding the collider analysis, a very basic question: how hard are the photons required to be? I do not see this information in (3.1) nor find it in the text. It would be useful to discuss how such a cut helps to reduce the background.

— Using the ingredients in Eq.(2.2) and (4.5) I estimate that the ratio between the cross section for $\eta$ pair production mediated by the contact top vertex and the one mediated by an off-shell Higgs scales as the fourth power of $(v m_\eta/fm_h)$. Do the authors agree? If so, it would be very useful to have this rough estimate in the text. The regime of interest (light $\eta$ and large $f$) is immediately identified.

— Unfortunately, the above ratio becomes of order unity for the models of Sec. 4, see (4.10), unless one fine-tunes the parameters of the theory such that the mass of $\eta$ is much smaller (as might happen in a couple of models discussed in sec 4). Fine-tuning is necessary if we want to have an unsuppressed $\eta^2 h$ coupling and simultaneously a small $\eta$ mass. Please comment on this important aspect in the text.
  • validity: -
  • significance: -
  • originality: -
  • clarity: -
  • formatting: -
  • grammar: -

Author:  Diogo Buarque Franzosi  on 2020-09-11  [id 958]

(in reply to Report 2 on 2020-09-05)

Thank you for your comments.
We addressed your questions below and modified the paper as indicated.

—Q: In the Introduction: why is the coupling of eta to gamma-gamma small/absent? Please refer to an explicit model or postpone an explanation of how/why this happens to Section 4.
A: A (CP-odd) coupling to the photons would be generated in the underlying gauge theory by the anomaly term. However, this is absent in the models under consideration [15]. Top quark loops do not contribute either (both CP) for the reason explained in Section 4, although this requires the selection of the spurions discussed in the text. We added an explanatory footnote in the introduction. This also answers the next question.

—Q: Also, why isn'’t there a CP-odd coupling to gluons? How much do your results depend on the absence of it?
A: The answer is essentially the same as for the previous question. The vanishing of the anomaly coefficient is generic for all these models and the absence of the eta t tbar coupling suppresses both eta g g and eta gamma gamma. See same footnote in the introduction.
If the gluon coupling was present, the phenomenology would be radically different indeed, since the single eta production would have a large cross-section at hadron colliders.

—Q: The sentence “.. via doubly suppressed anomalous interactions in a loop…” is rather unclear, especially if what the authors mean is essentially what is explained about the eta->bb rate when discussion Fig 3.
A: Yes, it is the same explanation. By “doubly suppressed” we meant that the anomalous eta VV coupling is suppressed by 1/f and, on top of it there is a loop suppression. But we agree that the wording is confusing and changed the sentence.

—Q: Still in the Introduction, in the paragraph below (1.1) it is said that “in models of PC this assertion needs to be better qualified”. Please anticipate that this is done in a later section, if this is the case.
A: We added the sentence "This will be discussed in detail in Section 4." This is a rather important but technical point, so we felt like mentioning it in the introduction, leaving the details for later.

—Q: Regarding the collider analysis, a very basic question: how hard are the photons required to be? I do not see this information in (3.1) nor find it in the text. It would be useful to discuss how such a cut helps to reduce the background.
A: The minimum pT is 10 GeV. Other than that, we used the standard object definition of Delphes with CMS card. We added a footnote in page 7 with some basic numbers including the minimum pt of the photons. Requiring harder photons worsens the sensitivity and we also added a sentence at the end of page 8 to clarify that.

—Q: Using the ingredients in Eq.(2.2) and (4.5) I estimate that the ratio between the cross section for eta par production mediated by the contact top vertex and the one mediated by an off-shell Higgs scales as the fourth power of (vm meta/f mh). Do the authors agree? If so, it would be very useful to have this rough estimate in the text. The regime of interest (light ? and large f) is immediately identified.
A: Putting $p^2=4 m_\eta^2$ in the Higgs propagator we find the ratio to be proportional to $(v/f m_h)^4 (4 m_\eta^2 - m_h^2)^2$ and not quite $(v m_\eta/f m_h)^4$ unless we are misunderstanding the question. The ratio goes to zero when the Higgs goes on-shell but indeed it has the same qualitative trend as you say for $m_\eta > m_h/2$. We would rather not put this formula in the paper though, since we did a full numerical treatment of the interference, momentum dependence and form factors for these contributions in section 4.

Q:— Unfortunately, the above ratio becomes of order unity for the models of Sec. 4, see (4.10), unless one fine-tunes the parameters of the theory such that the mass of eta is much smaller (as might happen in a couple of models discussed in sec 4). Fine-tuning is necessary if we want to have an unsuppressed $\eta^2 h$ coupling and simultaneously a small eta mass. Please comment on this important aspect in the text.
A: This is indeed an important point. We need a cancellation between the two terms on the RH side of the last two formulas in (4.10). These are the only two scenarios that allow us to have a small eta mass. We discuss this in the very last paragraph of section 4.

---

## Round 3 · Referee Report · Anonymous (Referee 1) · 2020-9-25

Report

The authors address all mz comments: I reccomend the manuscript for publication.

---

## Round 3 · Author Response

We addressed the referees' comments with minor text editing.

---

## Round 3 · List of Changes

• Last two sentences on 3rd paragraph of introduction about eta-fermion coupling.
  • Footnote on 4th paragraph of introduction about eta-photon and eta-gluon coupling.
  • Sentence on 4th paragraph of intro to clarify the smallness of fermion decay channel.
  • Reference to sec. 4 on paragraph 6 of intro for t-tbar-eta-eta detail.
  • Footnote about Delphes object definition and pt of photon on 1st paragraph of sec. 3.
  • Sentence about implication of photon pt on 6th paragraph of Sec. 3.1.
  • Further discussion about (3.3) just below it.
  • A parenthesis in conclusion specifying that the exclusion for large f is also only for small m_eta

---

## Editorial Decision

published